# The impact of the early COVID-19 pandemic on maternal mental health during pregnancy and postpartum

Catharina Bartmann[1]*, Theresa Kimmel[1], Petra Davidova[2,3], Miriam Kalok[2], Corina Essel[2], Fadia Ben Ahmed[2], Rhiannon V. McNeill[4], Tanja Wolfgang[1,4], Andreas Reif[2], Franz Bahlmann[5], Achim Wöckel[1], Patricia Trautmann-Villalba[6], Ulrike Kämmerer[1], Sarah Kittel-Schneider[2,4,7]

1 Department of Obstetrics and Gynecology, University Hospital, Würzburg, Germany, 2 Department of Psychiatry, Psychosomatic Medicine and Psychotherapy, University Hospital Frankfurt, Goethe University, Frankfurt am Main, Germany, 3 Department of Ophthalmology, University Hospital Frankfurt, Goethe University, Frankfurt am Main, Germany, 4 Department of Psychiatry, Psychosomatics and Psychotherapy, Center of Mental Health, University Hospital, University of Würzburg, Würzburg, Germany, 5 Department of Obstetrics and Gynecology, Buergerhospital Frankfurt, Frankfurt/Main, Germany, 6 Institute of Peripartal Interventions, Frankfurt am Main, Germany, 7 Department of Psychiatry and Neurobehavioural Science, University College Cork, Acute Mental Health Unit, Cork University Hospital, Wilton, Cork, Ireland

* Bartmann_c@ukw.de

## Abstract

### Purpose

The aim of this study was to investigate the effects of the COVID-19 pandemic on maternal mental health during pregnancy and the postpartum period.

### Methods

The impact of the COVID-19 pandemic situation during and post pregnancy was addressed on three main factors; maternal mental health, mother-child bonding, and maternal self-confidence. To do this, two different patient cohorts were compared; data from one cohort was collected pre-pandemic, and data was collected from the other cohort at the beginning of the pandemic. Questionnaires were used to collect data regarding depressive symptoms (Edinburgh Postnatal Depression Scale [EPDS]), anxiety (State Trait Anxiety Inventory [STAI]), maternal self-confidence (Lips Maternal Self-Confidence Scale [LMSCS]) and mother-child bonding (Postpartum Bonding Questionnaire [PBQ]).

### Results

There were no significant differences in depressive symptoms (EPDS with an average median of 4.00–5.00) or anxiety (STAI with an average median of 29.00–33.00) between the cohorts. However, the quality of postpartum maternal bonding was higher at 3–6 months in the pandemic cohort, which was also influenced by education and the mode and number of births. The maternal self-confidence was lower in the pandemic sample, also depending on the mode of birth delivery.

**Data Availability Statement:** All relevant data are within the manuscript and its Supporting Information files.

**Funding:** The author(s) received no specific funding for this work.

**Competing interests:** The authors have declared that no competing interests exist.

## Conclusions

In this study, a differential effect of the COVID-19 pandemic on mother-child bonding and maternal self-confidence was observed. The results thereby identified possible protective factors of the pandemic, which could potentially be implemented to improve maternal mental health and bonding to the child under normal circumstances.

## Introduction

The onset of the coronavirus induced disease 2019 (COVID-19) pandemic resulted in severe restrictions to personal freedom and had a major global impact on many areas of society, the economy, and health. The causative pathogen of COVID-19 is the coronavirus SARS-CoV-2 [1–3]. At the start of the pandemic, potential negative health effects of SARS-CoV-2 on pregnancies were unclear, and previous data were only available from the SARS- and MERS-coronaviruses. The latter were associated with a higher risk of adverse maternal and fetal pregnancy outcomes, such as severe maternal illness, miscarriage, and preterm birth [4,5]. After more than three years of circulating SARS-CoV-2, it is now known that direct transmission of SARS-CoV2 from mother to fetus is possible, but very rare [6,7]. However, a COVID-19 diagnosis during pregnancy has been shown as an additional risk factor for negative pregnancy outcomes, increasing the risk for caesarean delivery, preterm birth, pregnancy-related hypertensive disorders, fetal distress, and severe neonatal morbidity and mortality. There is also an increased risk for maternal mortality and ICU (intensive care unit) admission for women with a COVID-19 diagnosis [8–11].

Due to the potential risks of COVID-19 in general, numerous measures were implemented to protect against infection and limit the spread of SARS-CoV-2, especially at the beginning of the pandemic. These measures included "lock downs", kindergarten and school closures, restrictions of social contacts and movements, the obligation of testing for SARS-CoV-2, and the wearing of masks [2,3,12]. In case of a COVID-19 diagnosis, patients in the hospital or other medical facilities were isolated, with no or only limited visitors allowed [13]. There were also concerns regarding adequate medical care, for example treatment in an intensive care unit was no longer possible in some areas or countries due to the overloading of the health system [14–16]. However, the pandemic situation also revealed new opportunities; for example, 'home office' (working from home), along with video conferencing and virtual trainings, became possible and common for most workplaces [17,18].

Even under normal circumstances, pregnancy and postpartum are suggested to lead to a higher vulnerability for the first onset of mental health problems such as depressive and anxiety disorders. This might be due to the physical, psychological, and hormonal changes, as well as alterations in everyday life and work [19,20]. Approximately 10–15% of women develop a clinically relevant postpartum depression during this vulnerable phase. This has an impact on maternal self-efficacy and mother-child bonding, and therefore also on child development [21–23]. In 2020, multiple studies already began to investigate the effect of the COVID-19 pandemic and its associated burdens on women's mental health during pregnancy and postpartum. In spring 2020, an increase in depressive symptoms was already observed in pregnant women in China, which was specifically associated with the dynamics of COVID-19 infection numbers [24]. This initial finding was later supported by the results of various meta-analyses and reviews [25–28].

The aim of this study was to further determine how the COVID-19 pandemic may have impacted maternal mental health. To do this, data obtained during pregnancy and postpartum

from two cohorts was compared; one cohort had data collected prior to the COVID-19 pandemic situation (VBS study) [29] and one cohort had data obtained during the COVID-19 pandemic (GeZeCO study) [30]. Aspects such as depressive symptoms, anxiety, maternal self-efficacy and mother-child bonding were investigated, and significant differences between the cohorts were determined.

## Materials and methods

### Participant cohorts

**VBS cohort.** Participants of the VBS study (*Vater-Bindungs-Studie*—paternal bonding study) were mainly recruited in Frankfurt am Main, Germany, at the information evenings for expecting couples at the Department of Obstetrics in the Hospital "Bürgerhospital". Couples were included in the study during mid pregnancy ($> = 20$ weeks of gestation) between July 2017 and July 2018, altogether 84 women participated. The time points at which data were collected were during pregnancy (between 20 gestational weeks and directly before birth), 3 months postpartum, 6 months postpartum and 12 months postpartum. The follow-up visits were either performed at a study appointment in the hospital or at the family's home, if requested. The follow-up at 12 month was done over telephone and the self-report study questionnaires were filled in online at all study visits. The study procedure was previously published [29]. Data obtained from male partners were excluded for the purposes of this secondary study.

**GeZeCO cohort.** The GeZeCO study (*Geburt in Zeiten von COVID-19*—delivery in times of COVID-19) included 94 women who gave birth at the Department of Obstetrics and Gynaecology, University Hospital of Würzburg, Würzburg, Germany, during April to June 2020. The time points at which data were collected were during admission for birth, 3–6 months postpartum and 12 months postpartum. During the follow-up surveys, questionnaires were sent by postal services and the participants were called if there were no answers. None of the participants received a SARS-CoV-2 diagnosis during the study period. The study procedure was previously published [30].

**Ethics statement.** The VBS study was approved by the Ethics Committee of the University of Frankfurt and the Ethics Committee of the Hesse State Medical Association (Hessische Landesärztekammer) (approval no 135/17). The GeZeCO study was approved by the ethics committee of the University of Würzburg (No. 70/20 Amendment). After verbal and written information, the all participants agreed to participate in the study with written informed consent. Both studies adhered to the Declaration of Helsinki, version 2008 and 2013. The participants allowed the use for data for secondary analysis.

Fig 1 provides an overview of the procedures and structures of both studies, the maternal questionnaires used, and the defined comparison timepoints T1, T2a, T2b and T3 (Fig 1). Both studies were conducted in large hospitals with maximum obstetric care. Würzburg and Frankfurt are both located in the south of Germany and the distance between the two cities is ~120 km. The participants of both cohorts were therefore comparable regarding socio-economical and ethnic background and received care from the German health system with compulsory health insurance. Both study cohorts consisted predominantly of mentally healthy females.

### Questionnaires

**Depression and anxiety.** The Edinburgh Postnatal Depression Scale (EPDS) is a common global screening questionnaire for postnatal depression and includes ten items with a four-point Likert scale (from 0 to 3). The evaluation is based on a total sum score of all 10 items

| GeZeCO study (pandemic) | VBS study (pre-pandemic) |
|---|---|

**T 1**

| | |
|---|---|
| During delivery<br>(Date of inclusion:<br>April 16, 2020 to June 10, 2020)<br><br>94 participants<br><br>Edinburgh Postnatal Depression Scale<br>(EPDS) | 20 weeks of pregnancy to birth<br>(Date of inclusion:<br>July 12, 2017 to July 27, 2018)<br><br>84 participants<br><br>Edinburgh Postnatal Depression Scale<br>(EPDS) |

**T 2a**

| | |
|---|---|
| Follow-up 3-6 months after birth<br><br>62 participants<br><br>EDPS<br>State Trait Anxiety Inventory (STAI) -<br>Trait Scale<br>Postpartum Bonding Questionnaire<br>(PBQ)<br>Lips Maternal Self-Confidence Scale<br>(LMSCS) | Follow-up 3 months after birth<br><br>74 participants<br><br>EDPS<br>State Trait Anxiety Inventory (STAI) -<br>Trait Scale<br>Postpartum Bonding Questionnaire<br>(PBQ)<br>Lips Maternal Self-Confidence Scale<br>(LMSCS) |

**T 2b**

| | |
|---|---|
| | Follow-up 6 months after birth<br><br>80 participiants<br><br>EDPS<br>STAI<br>PBQ<br>LMSCS |

**T 3**

| | |
|---|---|
| Follow-up 12 months after birth<br><br>46 participants<br><br>EDPS<br>STAI<br>PBQ<br>LMSCS | Follow-up 12 months after birth<br><br>54 participants<br><br>EDPS<br>STAI<br>PBQ<br>LMSCS |

**Fig 1. Describes the structure of the VBS study and the GeZeCO study.**

[31]. There is a middle to high probability for depression if the total score of the EPDS has a value of 10 or higher [32]. There is a well-established German translation of the EPDS which has been in use for over 20 years [33–35].

The State Trait Anxiety Inventory (STAI) measures current (state scale) and persistent (trait scale) anxiety in two subscales. Each subscale consists of 20 items with a four-point Likert scale with a sum scale of 20 (low anxiety) to 80 (maximum anxiety intensity) [36]. The STAI was used to assess postpartum anxiety [37], via a well-established German translation [38]. In this study, the STAI Trait subscale (STAI Trait) was used for data analysis.

**Postpartum bonding and maternal self-confidence.** The Postpartum Bonding Questionnaire (PBQ) consists of 25 items with a six-point Likert scale (0–5). The total score can be formed by the sum of all items which must be partially inverted. A higher score indicates a worse mother-child-bonding [39,40].

The Lips Maternal Self-Confidence Scale (LMSCS) was developed to assess maternal self-confidence. The questionnaire contains 24 questions with a six-point Likert scale (1–6). For the evaluation, a sum score is formed, with a higher score associated with higher maternal self-efficacy [41].

## Statistical analysis and graphs

The software IBM SPSS Statistics 28.0 was used to create tables and perform statistical analysis. Normal distribution was determined by the Shapiro-Wilk-Test. The values are presented as mean with standard deviation (SD) or as median with the interquartile range. The Mann-Whitney-U test and the Pearson's chi-squared test were performed to determine significant differences. For correlations we used Spearman-rho test. P-values (p) $\leq$ 0.05 were classified as statistically significant. Graphs (box and whisker blots) were created using GraphPad Prism 9.5.1.

## Results

### Cohort demographics

The mean age of all participants was 33.65 ($\pm$4.16) years. The 94 participants of the GeZeCO study were statistically significantly younger (32.48$\pm$4.31 years) compared to the 84 VBS study participants (34.95$\pm$3.58 years; p<0.001). There was also a significant difference between the median BMI of participants in the GeZeCO study (23 kg/m$^2$) and in the VBS study (26.01 kg/m$^2$; p<0.001). Further characteristics of the study populations are given in Table 1. There were no significant differences in with regards to previous mental disorder or multiples. However, there were significant differences in degree of education (p< 0.001), mode of birth delivery (p = 0.005) and number of previous births (p< 0.001).

### Depression and anxiety

There was no significant difference in depressive symptoms between the cohorts at all time points (Table 2). Comparing the number of mothers with an EPDS sum score $\geq$ 10 and < 10, there were also no significant differences at T1 (p = 0.890), T2a (p = 0.086), T2b (p = 0.754) and T3 (p = 0.863).

The median score of the STAI Trait scale was nominally higher at T2a in mothers of the GeZeCO study (33.00 [29.00–37.00]) compared with participants of the VBS study (30.50 [26.00–39.00]). The same trend was observed at T2b (30.00 [25.00–36.00] vs. 33.00 [29.00–37.00]) and at T3 (29.00 [26.00–38.00] vs. 33.00 [30.00–38.00]). However, these differences were not statistically significant (Table 2).

**Table 1. Demographic and clinical characteristics of the study samples are shown.** Statistical differences are calculated with Pearson's Chi squared test and the level of significant difference was set at p = < 0.05. Statistically significant differences are displayed in bold.

| | | GeZeCO study | VBS study | Total number (line) | P-value (Pearson's chi-squared test) |
|---|---|---|---|---|---|
| Education | Non-academics | 59 | 25 | 84 | **< 0.001** |
| | Academic degree | 35 | 54 | 89 | |
| | Total number (row) | 94 | 79 | 173 | |
| Previous mental disorder | No | 87 | 70 | 156 | 0.183 |
| | Yes | 7 | 2 | 9 | |
| | Total number (row) | 93 | 84 | 177 | |
| Mode of birth delivery | Vaginal birth | 62 | 51 | 113 | **0.005** |
| | Vacuum extraction | 3 | 0 | 3 | |
| | Scheduled caeserian section | 20 | 4 | 24 | |
| | Unplanned/emergency caeserian section | 9 | 15 | 24 | |
| | Total number (row) | 94 | 70 | 164 | |
| Multiples | No | 87 | 82 | 169 | 0.124 |
| | Yes | 7 | 2 | 9 | |
| | Total number (row) | 94 | 84 | 178 | |
| Number of births | One | 49 | 63 | 113 | **< 0.001** |
| | More than one | 45 | 16 | 60 | |
| | Total number (row) | 94 | 79 | 173 | |

## Postpartum bonding

Maternal bonding quality was significantly better at T2a in the participants of the GeZeCO study than of those in the VBS study (6.00 [3.00–9.00] vs. 11.00 [6.00–14.00]; p <0.001). However, this was not true for the other time points (Table 3). There were significant correlations between the PBQ total score at T2a and education ($r_s$ = 0.212; p = 0.019), mode of delivery ($r_s$ = 0.233; p = 0.011), and number of previous births ($r_s$ = -0.278; p = 0.002). At T2a, significant correlations of PBQ total score with age ($r_s$ = 0.000; p = 0.996), BMI ($r_s$ = -0.033; p = 0.719), previous mental disorder ($r_s$ = 0.031; p = 0.742) as well as multiples ($r_s$ = -0.059; p = 0.520) could not be detected.

A detailed comparison at T2a between subgroups of the samples in relation to sociodemographic and obstetric covariables was performed. Quality of maternal bonding of participants with an academic degree was significantly higher in the GeZeCO study than in the VBS study (7.00 [4.00–9.00] vs. 11.00 [6.00–14.00; p = 0.014]. Individuals with a non-academic background showed a nominal, but not significant difference (6.00 [3.00–9.50] vs. 9.00 [5.00–

**Table 2. Presents the results of the EPDS and the STAI Trait scale with median and interquartile range at the different times T1, T2a, T2b and T3.**

| Sum scores | GeZeCO study | | VBS study | | p (Mann-Whitney-U) |
|---|---|---|---|---|---|
| | Median | Interquartile range | Median | Interquartile range | |
| EPDS T1 | 5.00 | 2.00–7.00 | 4.00 | 1.00–7.00 | 0.234 |
| EPDS T2a | 4.00 | 2.00–7.00 | 5.00 | 2.00–8.00 | 0.297 |
| EPDS T2b | 4.00 | 2.00–7.00 | 4.00 | 2.00–6.00 | 0.715 |
| EPDS T3 | 4.00 | 1.00–5.00 | 4.00 | 1.00–8.00 | 0.836 |
| STAI Trait T2a | 33.00 | 29.00–37.00 | 30.50 | 26.00–39.00 | 0.138 |
| STAI Trait T2b | 33.00 | 29.00–37.00 | 30.00 | 25.00–36.00 | 0.064 |
| STAI Trait T3 | 33.00 | 30.00–38.00 | 29.00 | 26.00–38.00 | 0.142 |

**Table 3. Presents the results of the PBQ and the LMSCS with median and interquartile range at the different times T2a, T2b and T3.**

| Total scores | GeZeCO study | | VBS study | | P-Value (Mann–Whitney-U) |
|---|---|---|---|---|---|
| | Median | Interquartile range | Median | Interquartile range | |
| PBQ T2a | 6.00 | 3.00–9.00 | 11.00 | 6.00–14.00 | <0.001 |
| PBQ T2b | 6.00 | 3.00–9.00 | 7.00 | 3.00–12.00 | 0.313 |
| PBQ T3 | 8.00 | 4.00–12.00 | 7.00 | 4.00–15.00 | 0.988 |
| LMSCS T2a | 119.00 | 110.00–124.00 | 123.00 | 112.00–130.00 | 0.099 |
| LMSCS T2b | 119.00 | 110.00–124.00 | 124.00 | 118.00–131.00 | 0.008 |
| LMSCS T3 | 125.00 | 117.00–131.00 | 126.00 | 117.00–135.00 | 0.531 |

12.00]; p = 0.106). A significant increase of maternal bonding quality in mothers who had a vaginal birth in both cohorts (p = 0.002) was found. Here, the maternal bonding was significantly higher (with lower values) in the GeZeCO study than in the VBS study (6.00 [3.00–9.00] vs. 9.50 [6.00–13.00]). This could be observed nominally, but not significantly for participants with planned caesarean section (6.00 [1.50–9.50] vs. 14.50 [10.00–21.00]; p = 0.058) and could not be measured for participants with unplanned/emergency caesarean section (15.00 [12.00–18.00] vs. 14.00 [8.50–20.00]; p = 0.879). In case of number of previous births, in both subgroups (one birth/more than one birth) there was a significant higher quality of maternal bonding (p = 0.046/p = 0.018) in the GeZeCO study than in the VBS study (7.00 [4.00–11.00] vs. 11.00 [6.00–14.00]/4.50 [3.00–7.00] vs. 8.00 [5.00–13.00]) (Fig 2).

## Maternal self-confidence

Maternal self-confidence was nominally lower in the GeZeCO study than in the VBS study at T2a (119.00 [110.00–124.00] vs. 123.00 [112.00–130.00]; p = 0.099) and significantly lower at T2b (119.00 [110.00–124.00] vs. 124.00 [118.00–131.00]; p = 0.008). There was a significant correlation between maternal self-confidence and mode of delivery at T2a/T2b ($r_s$ = -0.179; p = 0.049/$r_s$ = -0.250; p = 0.008), but not with age ($r_s$ = 0.005; p = 0.952/$r_s$ = -0.058; p = 0.537), BMI ($r_s$ = 0.086; p = 0.350/$r_s$ = 0.162; p = 0.089), education ($r_s$ = -0.014; p = 0.880/$r_s$ = 0.042; p = 0.658), previous mental disorders ($r_s$ = -0.025; p = 0.793/$r_s$ = -0.106; p = 0.283), multiples ($r_s$ = -0.064; p = 0.483/$r_s$ = -0.076; p = 0.420) and number of previous births ($r_s$ = 0.000; p = 1.000/$r_s$ = -0.032; p = 0.732).

By detailed comparison of the subgroups of the mode of delivery, the maternal self-confidence of patients with vaginal birth at T2a/T2b was significantly (p = 0.040/p = 0.001) lower in the GeZeCO study than in the VBS study (120.00 [113.00–124.00] vs. 124.00 [118.00–131.00]/120.00 [113.00–124.00] vs. 127.50 [119.50–134.00]. At T2a, this could be observed on a nominal level, but was not statistically significant for participants with scheduled caesarean section (119.00 [116.00–127.00] vs. 122.50 [114.00–126.00]; p = 0.851) and participants with unplanned/emergency caeserian section (107.00 [101.00–110.00] vs. 116.00 [103.00–125.00]; p = 0.503). At T2b, no difference between participants with planned caesarean section (119.00 [116.00–127.00] vs. 120.00 [119.00–124.00]; p = 0.769) and with unplanned/emergency caeserian section (107.00 [101.00–110.00] vs. 115.50 [107.00–125.00]; p = 0.234) was measured (Fig 3).

## Discussion

Pregnancy, childbirth, and postpartum are major life events leading to substantial physical, psychological and social changes [42–45]. So, it is not surprising that mental health problems such as depressive and anxiety disorder might occur more frequently for the first time during

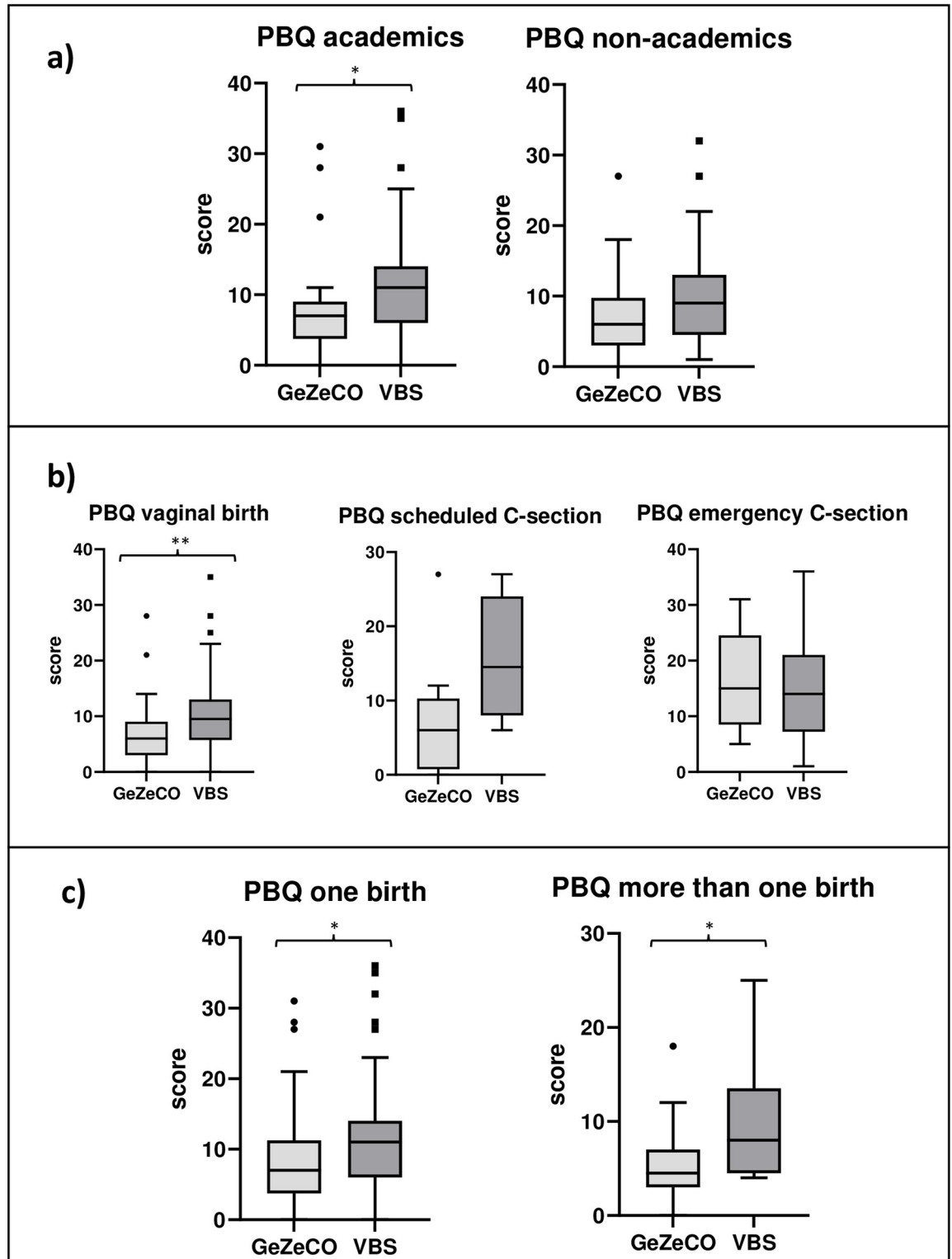

**Fig 2. Box and whisker plots with a detailed comparison of PBQ results of the subgroups at T2a.** Significant differences were marked (**p < 0.01; *p < 0.05).

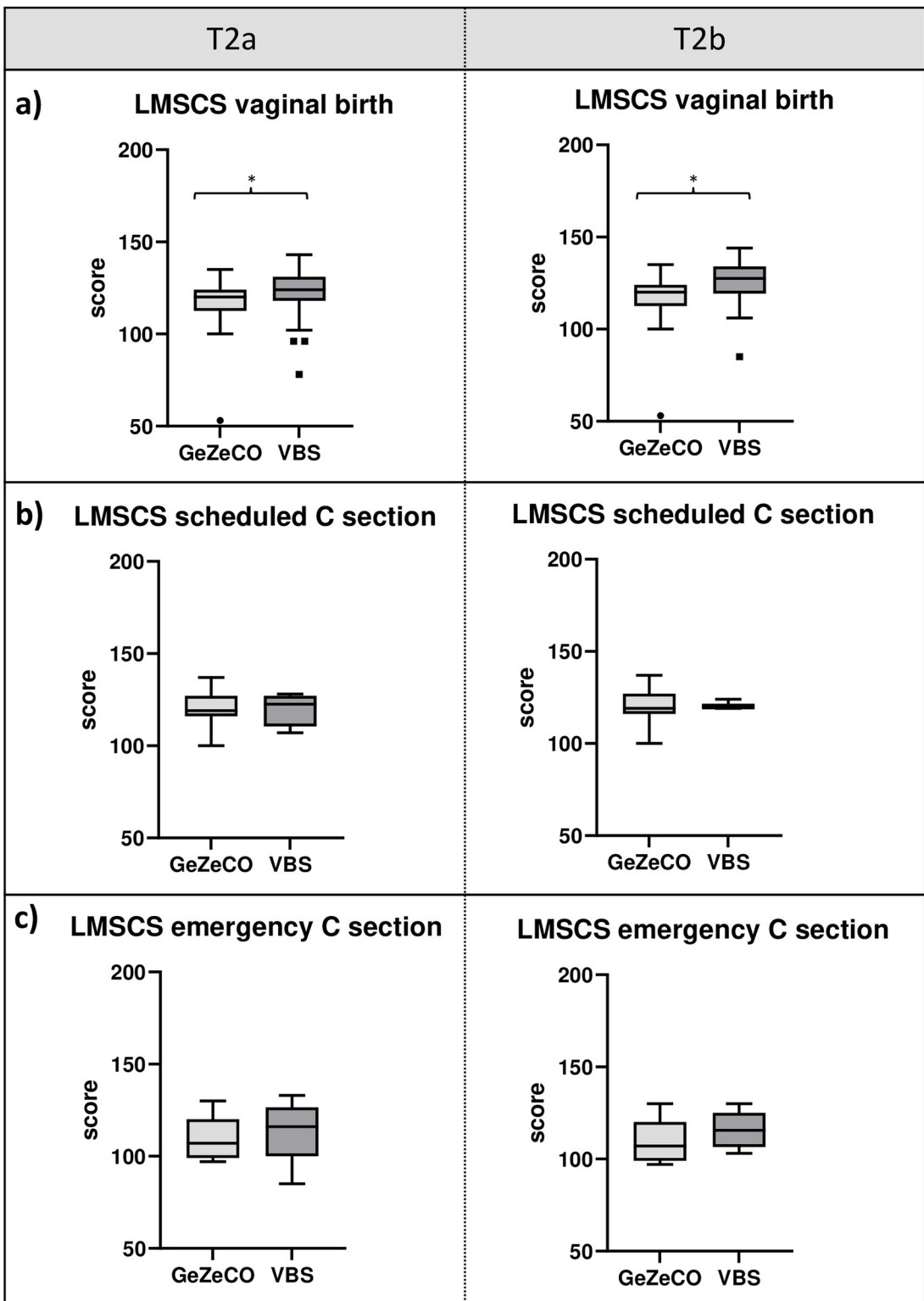

**Fig 3. Box and whisker plots with a detailed comparison of the LMSCS results of the subgroups at T2a and T2b.** Significant differences were marked (**p < 0.01; *p < 0.05).

this vulnerable period [19,20]. Considering the drastic ongoing experience of the COVID-19 pandemic and the consequent measures [1–3], the aim of this study was to compare measures of women's mental health during pregnancy and postpartum from a pre-pandemic period (VBS study) [29] with data from women at the beginning of the pandemic (GeZeCO study) [30].

Several recent studies found an increase in prevalence of peripartum depression and anxiety or depressive/anxiety symptoms in women during pregnancy and postpartum during the pandemic [24,46–56]. Meta-analyses and reviews further supported these findings [25–28]. However, in this study no significant changes in depressive or anxiety symptoms between the pre-pandemic and the pandemic study cohort were found, similar to other recent large Dutch and Chinese studies [57,58]. This discrepancy in results could be due to the smaller sample size used in the GeZeCO and VBS study, and the different time points at which data was collected. Additionally, in both studies, the participants were personally assessed by study staff whereas other previous studies incorporated online recruitment and/or online surveys [46–50,52,54].

Another factor potentially influencing the effect of the pandemic on maternal mental health is the country in which the study was conducted, for example China [24,51], Canada [47], the UK [46,54,59], France [50], the USA [52], Italy [53], Iran [60] and Turkey [56]. These conflicting results could be due to differences in political and economic systems, COVID-19 containment measures, health insurance practices, and the general state of the healthcare system [61,62]. For example, one study from China during the first wave of the COVID-19 pandemic reported that the number of women receiving antenatal care decreased by ~50% [51]. Furthermore, whether women were examined during pregnancy or in the postpartum period also appears to affect the study outcome. For example, Scandura et al. only observed an increase in antepartum depression, but not in postpartum depression [53].

In a previous analysis an association between COVID-19-related fear and anxiety, and pandemic-related concerns, and depression was found [30]. Further studies specifically identified factors contributing to peripartum depression and anxiety. These include no or low social support, greater social distancing, a lower knowledge or education, increased workload, pregnancy pathologies, obesity, young children during the lockdown, friends/relatives with COVID-19 [48–50,63–67]. Possibly another reason for the lack of differences in depression and anxiety between the two cohorts are the relatively few cases of COVID-19 in Würzburg during the first wave of the COVID-19 pandemic compared to other countries for an overview of the whole pandemic in Bavaria see here [68].

Most interestingly, the results showed a higher maternal bonding quality at 3–6 months postpartum in the pandemic cohort. This result was potentially influenced by education (an academic degree), mode of delivery (vaginal birth), and number of previous births (one birth/more than one birth). In these subgroups of mothers, maternal bonding may have benefited from social distancing during the pandemic. Additionally, it has been reported that during the first postpartum period mental health might benefit from less external stimuli (less visitors after birth), more focusing on the new baby and on the core family [69,70]. Furthermore, the pandemic has also facilitated work flexibility, with the possibility of partial or complete home office workplace. This could also have positively influenced maternal mental health and mother-child bonding due to the partner being able to be more available and supportive [71,72].

In contrast to the observed improvement in mother-child-bonding in the pandemic cohort, the data suggested that maternal self-confidence was lower 3–6 months postpartum, potentially influenced by the mode of delivery (vaginal birth). In the analysis of the pre-pandemic GeZeCO study before the maternal self-confidence correlated negatively with the perceived impairments of the COVID-19 pandemic [30]. This result is consistent with recent studies. It

has been proposed that the mother's limited contact with other mothers could lead to lower self-efficacy [73]. The relationship between depression, anxiety, maternal self-efficacy, the mother-child bonding and the spectrum of circumstances caused by the COVID-19 pandemic is complex, with contrasting results published. For example, a study from the USA reported an association between low mother-child bonding and COVID-19-related grief, whereas another study reported an association between higher mother-child bonding and COVID-19-related health worries [74]. This effect of COVID-19-related fears and worries on maternal bonding was also observed in another study, which reported an association between increased anxiety and improved bonding [75]. In contrast, depressive symptoms were associated with reduced attachment to the infant or a lower maternal-fetal bonding during the pandemic [46,75].

## Limitations

One of the main limitations of this study is that the two used cohorts were not primarily designed for comparison with each other and were both designed to investigate different research questions. As a result, the data collection time points between the two cohorts varied slightly. Additionally, the cohorts significantly differed on several demographic factors, including age and BMI, and the cohorts were based in different locations in Germany. This also meant that the studies were performed at two different hospitals, although the hospitals were characteristically similar in their medical care systems. Lastly, the total number of participants included in the study is relatively small, and therefore the study may be underpowered. Furthermore, the pandemic cohort was only sampled at the beginning of the pandemic.

## Conclusions

In this study, no effect of the COVID-19 pandemic on depression or anxiety during pregnancy and the postpartum was observed. However, maternal self-efficacy appeared decreased, and the quality of mother-child bonding increased, both of which were influenced by demographic variables (such as education and mode of delivery). The results therefore suggest that the pandemic did not have an overall negative effect on maternal mental health but could negatively (maternal self-efficacy) and positively (mother-child bonding) influence individual aspects. Factors that were able to positively modulate aspects of maternal mental should be further investigated, to determine whether they could be routinely implicated in future peripartum care.

## Supporting information

**S1 Table. Summarized raw data of the questionnaires in the prepandemic and pandemic cohort.**
(PDF)

## Acknowledgments

We thank the staff in the Department of Obstetrics of the Bürgerhospital Frankfurt and the University Hospital Würzburg for support in recruitment of the participants and we thank all participants for being part in our studies.

## Author Contributions

**Conceptualization:** Catharina Bartmann, Theresa Kimmel, Ulrike Kämmerer, Sarah Kittel-Schneider.

**Data curation:** Theresa Kimmel, Petra Davidova, Miriam Kalok, Corina Essel, Fadia Ben Ahmed, Tanja Wolfgang, Andreas Reif, Franz Bahlmann, Patricia Trautmann-Villalba.

**Formal analysis:** Catharina Bartmann, Ulrike Kämmerer, Sarah Kittel-Schneider.

**Investigation:** Catharina Bartmann, Sarah Kittel-Schneider.

**Supervision:** Rhiannon V. McNeill.

**Writing – original draft:** Catharina Bartmann, Ulrike Kämmerer, Sarah Kittel-Schneider.

**Writing – review & editing:** Catharina Bartmann, Theresa Kimmel, Miriam Kalok, Corina Essel, Fadia Ben Ahmed, Rhiannon V. McNeill, Tanja Wolfgang, Andreas Reif, Franz Bahlmann, Achim Wöckel, Patricia Trautmann-Villalba, Ulrike Kämmerer, Sarah Kittel-Schneider.

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
