## [Decision Letter · Decision Letter 0]

19 Mar 2024

PONE-D-24-04080The impact of the early COVID-19 pandemic on maternal mental health during pregnancy and postpartumPLOS ONE

Dear Dr. Bartmann,

Thank you for submitting your manuscript to PLOS ONE. After careful consideration, we feel that it has merit but does not fully meet PLOS ONE’s publication criteria as it currently stands. Therefore, we invite you to submit a revised version of the manuscript that addresses the points raised during the review process.

We look forward to receiving your revised manuscript.

Kind regards,

Ikechukwu Innocent Mbachu

Academic Editor

PLOS ONE

Journal Requirements:

2. In the online submission form, you indicated that "The data that support the findings of this study are not openly available due to reasons of sensitivity and are available from the corresponding author upon reasonable request."

**Additional Editor Comments:**

The authors should respond to the reviewers' queries.

Reviewers' comments:

Reviewer's Responses to Questions

**Comments to the Author**

1. Is the manuscript technically sound, and do the data support the conclusions?

Reviewer #1: Yes

Reviewer #2: Yes

2. Has the statistical analysis been performed appropriately and rigorously? 

Reviewer #1: Yes

Reviewer #2: I Don't Know

3. Have the authors made all data underlying the findings in their manuscript fully available?

Reviewer #1: Yes

Reviewer #2: No

4. Is the manuscript presented in an intelligible fashion and written in standard English?

Reviewer #1: No

Reviewer #2: Yes

5. Review Comments to the Author

Reviewer #1: Introduction:

line 64, please rephrase line 64, in the first sentence to enhance understandability.

line 74: state full meaning of NICU .

Method:

State the means of the postpartum follow-ups, virtually or physically or mixed?

Reflect the sample size(s).

Result:

line 240-- grammar check. ... but not was not...

Conclusion:

Line 329, ... but could negatively and positively influence individual aspects. OF WHAT?

You mean aspects of self-confidence and bonding respectively? Please clarify.

Reference:

Please, manually edit and format most of the references in the reference page.

General manuscript presentation not well structured. E.g Seeing result table in the Methodology subsection.

Reviewer #2: The manuscript should be impersonal. It is a manuscript not a descriptive essay as the authors depicted in some parts of the manuscript. All pronouns such as, 'we' and our should be removed and replaced appropriately. This was shown in sections such as the abstract, last paragraph in introduction session, Post partum bonding, conclusion etc.

6. PLOS authors have the option to publish the peer review history of their article (what does this mean?). If published, this will include your full peer review and any attached files.

Reviewer #1: **Yes: **Dr. Sunday O Oriji

Reviewer #2: No

---

## [Author Response · Author response to Decision Letter 0]

17 Jun 2024

Dear Editor,

thank you for the appreciation of our work and we are happy to provide a minor revision of the manuscript.

We have addressed your additional requirements.

Done.

2. In the online submission form, you indicated that "The data that support the findings of this study are not openly available due to reasons of sensitivity and are available from the corresponding author upon reasonable request."

The participants have only consented to the anonymized publication of the summary statistics and therefore we are only able to share the raw pseudonymised data with scientific colleagues on request. This is because publication of the data sets could lead to the subjects being identified via the large amount of information we have collected.

We have included the full ethics statement in the ‘Methods’ section of our revised manuscript file.

We have reviewed our reference list and corrected reference 35 and 37: 

• [35] Spielberger CD, Gorsuch RL, Lushene RE. Manual for the State-Trait Anxiety Inventory. Palo Alto, CA, 1970: Consulting Psychologists Press 

• [37] Laux L, Glanzmann P, Schaffner P, Spielberger CD. Das State-Trait-Angstinventar. Theoretische Grundlagen und Handanweisung. Weinheim, 1981: Beltz Test GmbH

There was a rather minor erratum of the references which we have included in the reference list. We therefore have two more literature sources.

• [9] Villar et al. 2021: Misspelled Author Name in the Byline. JAMA Pediatr. 2022 Jan 1;176(1):104. doi: 10.1001/jamapediatrics.2021.4953. Erratum for: JAMA Pediatr. 2021 Aug 1;175(8):817-826. PMID: 34779828; PMCID: PMC8593825.

• [47] Lebel et al. 2020: Lebel C, MacKinnon A, Bagshawe M, Tomfohr-Madsen L, Giesbrecht G. Corrigendum to elevated depression and anxiety symptoms among pregnant individuals during the COVID-19 pandemic journal of affective disorders 277 (2020) 5-13. J Affect Disord. 2021 Jan 15;279:377-379. doi: 10.1016/j.jad.2020.10.012. Epub 2020 Nov 18. Erratum for: J Affect Disord. 2020 Dec 1;277:5-13. PMID: 33099052; PMCID: PMC8445275.

We have processed the reviewers' comments point by point and commented on them accordingly. We hope that our manuscript now fulfills the criteria for publication.

Thank you very much!

Catharina Bartmann

Response to Revieweers:

Reviewer #1: Introduction:

line 64, please rephrase line 64, in the first sentence to enhance understandability.

Many thanks for this hint. We have rephrased the two sentences as suggested.

line 74: state full meaning of NICU .

NICU means “neonatal intensive care unit”. We noticed by your comment that we had written “NICU” instead of “ICU” (intensive care unit). Since this sentence is about maternal risks, ICU is the correct term and we have explained the abbreviation.

Method:

State the means of the postpartum follow-ups, virtually or physically or mixed?

Thank you, we have added the requested information about the postpartum follow-ups in the methods section.

Reflect the sample size(s).

We have added the respected sample sized in the method section for clarification. 

Result:

line 240-- grammar check. ... but not was not...

Thanks, done.

Conclusion:

Line 329, ... but could negatively and positively influence individual aspects. OF WHAT?

You mean aspects of self-confidence and bonding respectively? Please clarify.

Thank you, we have clarified it in the revised conclusion.

Reference:

Please, manually edit and format most of the references in the reference page.

We have reviewed our reference list and corrected reference 35 and 37: 

• [35] Spielberger CD, Gorsuch RL, Lushene RE. Manual for the State-Trait Anxiety Inventory. Palo Alto, CA, 1970: Consulting Psychologists Press 

• [37] Laux L, Glanzmann P, Schaffner P, Spielberger CD. Das State-Trait-Angstinventar. Theoretische Grundlagen und Handanweisung. Weinheim, 1981: Beltz Test GmbH

There was a rather minor erratum of the references which we have included in the reference list. We therefore have two more literature sources.

• [9] Villar et al. 2021: Misspelled Author Name in the Byline. JAMA Pediatr. 2022 Jan 1;176(1):104. doi: 10.1001/jamapediatrics.2021.4953. Erratum for: JAMA Pediatr. 2021 Aug 1;175(8):817-826. PMID: 34779828; PMCID: PMC8593825.

• [47] Lebel et al. 2020: Lebel C, MacKinnon A, Bagshawe M, Tomfohr-Madsen L, Giesbrecht G. Corrigendum to elevated depression and anxiety symptoms among pregnant individuals during the COVID-19 pandemic journal of affective disorders 277 (2020) 5-13. J Affect Disord. 2021 Jan 15;279:377-379. doi: 10.1016/j.jad.2020.10.012. Epub 2020 Nov 18. Erratum for: J Affect Disord. 2020 Dec 1;277:5-13. PMID: 33099052; PMCID: PMC8445275.

General manuscript presentation not well structured. E.g Seeing result table in the Methodology subsection.

Many thanks for this hint. We have included Table 1 in the results section and carefully checked the rest of the structure of the manuscript.

Reviewer #2: The manuscript should be impersonal. It is a manuscript not a descriptive essay as the authors depicted in some parts of the manuscript. All pronouns such as, 'we' and our should be removed and replaced appropriately. This was shown in sections such as the abstract, last paragraph in introduction session, Post partum bonding, conclusion etc.

Thank you. We replaced the personal pronouns in whole manuscript.

---

## [Editor Report · Decision Letter 1]

9 Sep 2024

The impact of the early COVID-19 pandemic on maternal mental health during pregnancy and postpartum

PONE-D-24-04080R1

Dear Dr. Bartmann

We’re pleased to inform you that your manuscript has been judged scientifically suitable for publication and will be formally accepted for publication once it meets all outstanding technical requirements.

Kind regards,

Monia Marchetti

Academic Editor

PLOS ONE

Additional Editor Comments (optional):

The revised manuscript accomplished all the requested changes and is suitable for publication.
---

## [Editor Report · Acceptance letter]

12 Sep 2024

PONE-D-24-04080R1 

PLOS ONE

Dear Dr. Bartmann, 

I'm pleased to inform you that your manuscript has been deemed suitable for publication in PLOS ONE. Congratulations! Your manuscript is now being handed over to our production team.

Kind regards, 

on behalf of

Dr. Monia Marchetti 

Academic Editor

PLOS ONE